# R-SAC: Reinforcement Sample Consensus

**Zhaoyang Huang**
The Chinese University of Hong Kong
Shatin, Hong Kong
1155152331@cuhk.link.edu.hk

**Yan Xu**
The Chinese University of Hong Kong
Shatin, Hong Kong
yanxu@cuhk.link.edu.hk

## Abstract

The rejection of outliers in observed data is the foundation for accurate model estimation. Random sample consensus (RANSAC) is a classical algorithm aiming to find the inliers for robust model estimation. After sampling a series of minimal sets that can support the hypothesis estimation and generating the respective hypotheses, the best hypothesis that earns the maximum consensus is chosen for the final model estimation. However, this strategy may face exponentially computational growth as the outlier ratio increases. Besides, fitting a model from a minimal set may hinder the accurate model estimation especially when the inliers are extremely rare. In contrast, a model estimated from more observations may be better than from a minimum set. To approach such problem, we propose reinforcement sample consensus (R-SAC) to train a neural network to classify the inliers and outliers among all the correspondences with reinforcement learning. During training, we regard the number of inliers as a reward and encourage the agent to find the optimal subset supporting the final model estimation in a unsupervised manner. During inference, the R-SAC network is able to directly generate the inlier set, which could significantly reduce the computational resources in sampling and is able to select a more robust model hypothesis fitted from more correspondences. Empirical results show that our method achieves comparable performance compared with the previous supervised counterparts and remarkable efficiency especially when the outlier ratio is large.

## 1 Introduction

In computer vision, model estimation is of fundamental importance and is widely used in camera calibration, localization, registration etc. To obtain the optimal model supported by the given observation set, we generally devise appropriate error metrics serving as the objective function during model optimization. However, due to the non-ideal sensors or the algorithm imperfectness, noise inevitably exists in the observation set, which may brings some inherent errors and dramatically decrease the estimation accuracy. Thus recognizing and rejecting outliers is a pivotal procedure for model estimation. Random sample consensus (RANSAC) is a classic algorithm for outlier removal. From the view of statistics, the inliers tend to consistently support the same model while the outliers may diversely support some random models. Hence, the model computed from an outlier-free observation set should be supported by most of the inliers. With the assumption that the best model are supported by the majority of observations, RANSAC iteratively samples a subset from the whole observation set, so called the minimal set, to create model hypotheses with a minimum solver and choose the hypothesis that are supported by the most observations as the best model $h^*$. The observations with errors for $h^*$ greater than a certain threshold are deemed as outliers and the remaining observations are inliers. A minimum solver can compute the model parameters from minimal observations. For example, computing parameters of a line only requires two points. Supposing a model solver requires $k$ reliable observations and the actual inlier ratio is $r$, the probability $p$ of obtaining at least one outlier-free subset after sampling $n$ sub-sets can be derived

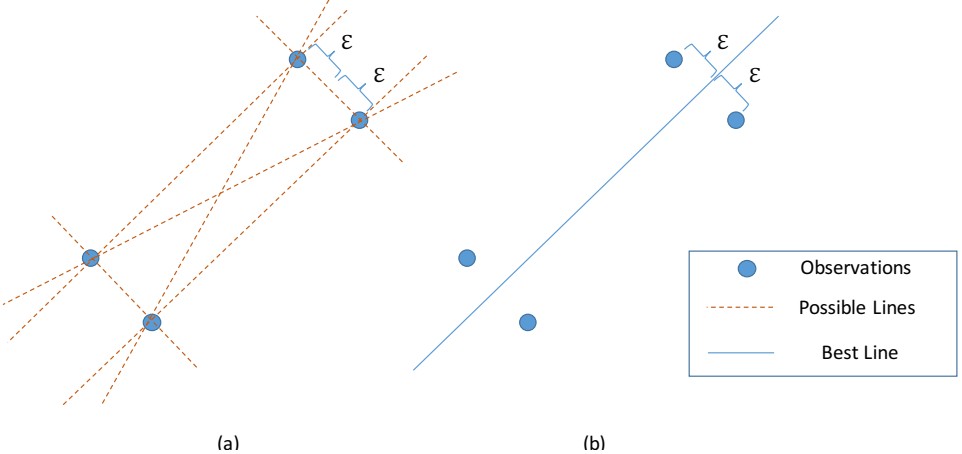

Figure 1: Line fitting example. (a) six possible solutions from the minimal solver of line fitting. (b) the best line that computed from all of the four points.

as:

$$p = 1 - (1 - r^k)^n \tag{1}$$

Apparently, a smaller sub-set cardinality $k$ can achieve higher success probability $p$ with the same inlier ratio $r$ and the sample number $n$. Therefore, RANSAC algorithm always use a minimal solver (requiring the minimal $k$) to fit the model during sampling. With the development of machine learning, some recent works () adopt neural networks to predict the sampling prior probability for guiding the sub-set sampling process in classic RANSAC algorithm to improve the efficiency. Thanks to the informative guidance, the convergence speed is largely improved. However, they still adopt the minimal solver which still limits the proposal space of model hypotheses, i.e., the hypothesis can only be computed from a minimal set, which might be biased and erroneously reject the inliers. Take the line fitting problem as an example as shown in Fig. 1. Suppose there are four points ( (Fig. 1)) and the inlier threshold is $\epsilon$. Apparently there are six possible solutions for the minimal solver as shown in Fig. 1-(a) since a line can be defined by two points, but all of these line hypotheses will reject the other two points with the current rejection threshold $\epsilon$ and thus achieves inferior results. However, the best line is actually estimated from all of the four points as shown in Fig. 1-(b), which unfortunately beyond the capacity of the minimal solver.

Recently, Moo et al. (11) train a neural network to directly select observations, which is able to generate a model from more observations. Nevertheless they requires costly ground truth labels for the inliers and outliers in the training stage. To get rid of labor-intensive annotation while still be able to estimate a model from as many reliable observations as possible. we propose reinforcement sample consensus (R-SAC) that learns an inlier selection policy $\pi_\theta(a_t|s_t)$ in a self-supervise manner. Unlike Moo et al. (11) directly supervise the inlier/outlier classification with the groundtruth labels, R-SAC adopts proximal policy optimization (PPO) to gradually learn the best strategy through iterative exploration and exploitation in an self-supervised manner. Given a set of observations (state $s_t$) and a inlier selecting action $a_t$, the environment provides two kinds of feedback: reward and the signal signifying whether the sampling procedure is done. We deem that a model hypothesis is better if it is supported by more observations and thus set the inlier amount of current model (estimated after action $a_t$) as the reward value for the state-action pair $(s_t, a_t)$. The environment will return done signal if the derived model can cover all observations because the current observation selection is an optimal solution. To this end, R-SAC trains an agent to predict the selection probability $\pi_\theta(a_t^{obs}|s_t)$ for each observation and a stop iteration probability $\pi_\theta(a_t^{stop}|s_t)$, which constitutes the overall policy $\pi_\theta(a_t|s_t)$. During training, R-SAC collects the selected observations as the state $s_{t+1}$ for the next iteration until the done signal is received.

In this way, the policy of observation selection is evaluated by the expectation of inlier number upon the predicted stop probability. The best policy is that the agent can select observations that maximizes the inlier number in the first step and stop the iteration.

## 2 Related Work

RANSAC has been widely applied in the field of 3D vision, such as object pose estimation (13) and visual localization (16). After Fischler and Bolles first introduced the basic RANSAC pipeline (5), many augmented variants are proposed to improve the efficiency especially when the outliers prevail. Universal RANSAC (USAC)(14) is one of the most successful RANSAC algorithms solely relying on handcrafted strategies. The algorithm includes sampling hypothesis by progressive guidance (PROSAC)(4), fitting the model with local optimization (LOSAC), and efficiently verifying hypothesis (ORSAC). Recently, machine learning has shown its great potential in capturing statistical information. Based on this, Brachmann et al. propose the Differentiable RANSAC (DSAC)(2; 1) to learn consistent correspondences for visual localization. More recently, NG-RANSAC (3) further devise a neural network to predict the sampling probabilities which guide the sampling process in RANSAC. With large amount of training data the sampling prior can be learned and modeled by the deep neural networks in a data driven manner, and the NG-RANSAC therefore consistently outperforms previous methods. However, they limit the model hypothesis space since they only takes the minimal samples to construct the hypotheses. Our proposed R-SAC further amends the sampling strategy during iteration so that we can pick out a better hypothesis with less steps.

## 3 Proposed Approach

### 3.1 Framework

The goal of RANSAC is to accurately distinguish inlier data for model estimation from a set of noisy observations $Y = \{y_i | i = 1, 2, 3, ...\}$. Suppose the target model can be estimated from at least $N$ observations which constitutes a basic set called minimum set, then for each sampled minimum set, we can obtain a model hypothesis $h$ with the predefined model estimator $f$: $h = f(y_1, y_2, ..., y_N)$. In the RANSAC pipeline, we usually randomly sample with which we fit the corresponding model hypotheses $M$ minimal sets $H = \{h_1, h_2, ..., h_M\}$, and then select the model with the highest score as the best hypothesis $h^*$. In practice, the inlier count for each model is generally served as the scoring function $s(h, Y)$ in the classic RANSAC algorithm and the above best hypothesis generation procedure can be formulated as

$$h^* = \arg \max_{h^* \in H} R(h^*, Y) \tag{2}$$

During sampling, the sampling probability $p(y_i)$ of different observations $y_i \in Y$ directly affect the model hypothesis generation results $H$. In the classic RANSAC algorithm, due to the absence of prior information, $p(y_i)$ is set to be equal for all the observations, i.e., a uniform sampling is adopted. Apparently, a sampling process with prior guidance could improve the model hypotheses quality compared with such blind sampling. NG-RANSAC (3) testifies that the neural network with parameter $\mathbf{w}$ has the capability to provide a reasonable distribution $p(y; \mathbf{w})$ to guide the sampling process in RANSAC algorithm.

In contrast with NG-RANSAC generating a model hypothesis from a minimum observation set, we predict the confidence for each observation and the model $h^*$ is derived from all the confident observations. To achieve this goal, we utilize PPO to explore and find the best policy to collect inlier observations for model estimation. As shown in Fig. 2, the actor network continuously output the policy prediction for the current step to pursue a larger reward (score) during training. Hence, the return of a trajectory can be expressed as:

$$G(s_0) = \sum_{t=0}^{T} \gamma^t R(s_t, a_t^{obs}) \tag{3}$$

Suppose the best solution at step $t$ is $a_t^{obs*}$, then we have $R(s_0, a_t^{obs*}) \geq R(s_t, a_t^{obs*}) > \gamma^t R(s_t, a_t^{obs*})$ because the observation $s_t$ is a subset of $s_0$ as shown in Fig. 2 and we take the inlier count as the reward function. To push the agent make the wise decision within less steps and avoid the overlong trajectories, we further design a network to predict the stop probability $a_i^{stop}$ for the current step. We take the predicted stop probability as the weight for the reward at each step. Concretely, for the reward $R(s_t, a_t^{obs})$ at step $t$, its weight is the probability that the agent should stop at this step: $\prod_{i=0}^{t-1}(1 - a_i^{stop})a_t^{stop}$. It is easy to find that the summation of all these weights

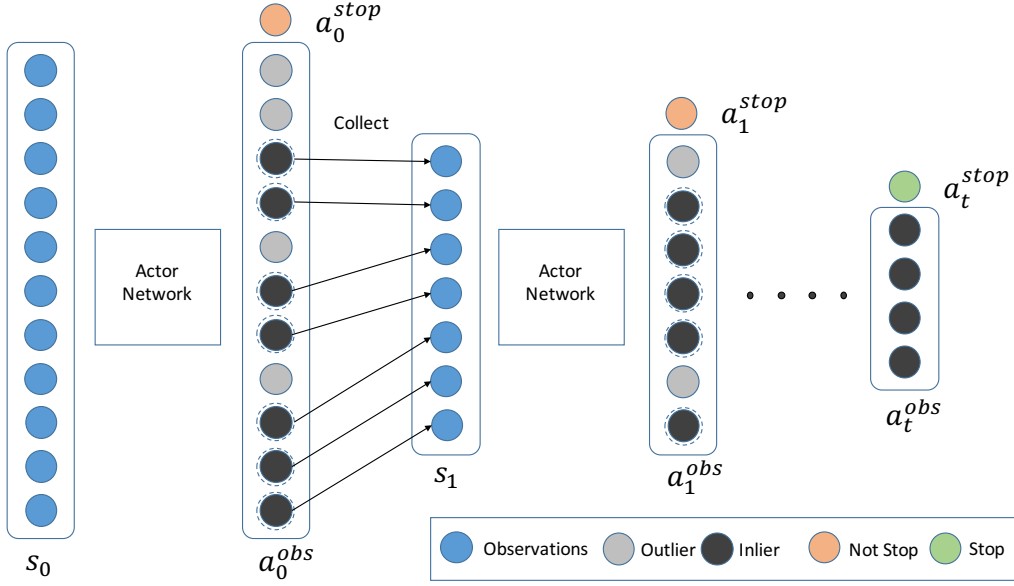

Figure 2: Reinforcement sample consensus.

along a trajectory equals to 1: $\sum_{t=0}^{T}\prod_{i=0}^{t-1}(1-a_i^{stop})a_t^{stop}=1$. Thus, the weighted version of return function can be rewritten as:

$$G(s_0)=\sum_{t=0}^{T}\prod_{i=0}^{t-1}(1-a_i^{stop})a_t^{stop}\gamma^t R(s_t,a_t^{obs}) \qquad (4)$$

In this manner, the agent could adjust the stop probability to tune the weights in unsupervised manner. Given that $G(s_t)=a_t^{stop}R(s_t,a_t^{obs})+\gamma(1-a_t^{stop})G(s_{t+1})$ and the best solution will be propagated to earlier steps, we have $R(s_t,a_t^{obs*})>\gamma^t G(s_{t+1})$, which means the agent tends to stop earlier for larger rewards. In this settings, the agent is encouraged to find the solution that takes larger rewards in earlier steps and shorten the iterations. In query stage, we run the R-SAC with 100 iteration and choose the model that covers the most inliers.

### 3.2 Environment Setup

**States.** In our environment, the state $s_t$ is the set of input observations. After the selecting action, the set of selected observations is the next state $s_{t+1}$.

**Reward.** The reward of the agent is defined as the score function $R(h_t,Y_t)$ in Eq. (2), which is practically set to be the inlier count with the model hypothesis derived from the selected observations $h_t$.

**Action.** At each step $t$, the agent need to output a selection mask $a_t^{obs}\in\mathbb{R}^K$ containing the selection indicator elements $a_t^{obs}(i)\in\{0,1\}$ where 0 indicates not to select and 1 represents to select. Besides, the agent also predicts a stop probability $a_t^{stop}$ to determine whether stop the iteration.

**Policy.** The policy function outputs the probabilities of different actions in the action space. In our situation, we utilize a neural network module to predict the selection confidence $p(y_i;\mathbf{w})$ of each observation and define the policy function as $\pi(a^{obs}|s)=\prod_i p(y_i;\mathbf{w})$ assuming an independent sampling of different observations. The stop probability $a^{stop}$ predicted by the agent denotes the confidence that whether we have gotten the best model.

**Temporal difference.** Based on the stop action, the Bellman optimality equation for the value function is:

$$v^\pi(s_t)=a_t^{stop}R(s_t,a_t^{obs})+\gamma(1-a_t^{stop})\sum_{s_{t+1}\in S}P(s_{t+1}|s_t,a_t^{obs})v^\pi(s_{t+1}),$$

and the corresponding temporal difference error is:

$$\delta_t = a_t^{stop} R(s_t, a_t^{obs}) + \gamma(1 - a_t^{stop})v^\pi(s_{t+1}) - v^\pi(s_t)$$

## 3.3 Deep Neural Networks

We use the neural network architecture proposed by Moo etal (11). The network is contructed by multiple residual blocks made of multi-layer preceptron (MLP), ReLU activation function (12), instance normalization (17), and batch normalization (9). The instance normalization transfers the feature information among all observations where all the calculation is permutation-invariant, which makes the overall neural network invariant to the permutation of observations. At the end of the network, a sigmoid function is applied, which outputs a confidence ranging from 0 to 1 for each observation.

To approximate the value function and predict a stop probability, we extend two branches at the end of the network. The two branches is also constructed by a residual block, a global average pooling, and a many-to-one fully connected layer, which outputs a scalar encoding the global information. The stop probability branch is additionally followed by a sigmoid function.

## 4 Experiments

We take the fundamental matrix estimation, a basic task in 3D vision, to evaluate our proposed R-SAC on.

**Fundamental Matrix Estimation.** In epipolar geometry, fundamental matrix is a $3 \times 3$ matrix denoting the relationship between two images in the same scene. Based on the fundamental matrix defined regarding two images, we can relate the correspondence points among two images with the epipolar geometry. Reversely, we can also estimate the fundamental matrix given enough point pair correspondences. Popular fundamental matrix estimation algorithms include the Eight-point algorithm (8), the Direct Linear Transformation (7), and the general least square fitting. The Eight-point algorithm is able to solve for the fundamental matrix with at least 8 pairs of correspondence, and NG-RANSAC therefore utilizes it as a minimal solver and sample eight correpondences in each iteration. For fair comparison, our experiment is also based on the Eight-point algorithm but we do not strictly limit the number of correspondence used in fundamental matrix solving as mentioned in Sec 3.

**Dataset.** We compare with previous methods by taking subsequent image pairs within sequences in the KITTI dataset (6) . Following Deep F (15), we train R-SAC on 00-05 sequences and test on 06-10 sequences. We extract SIFT (10) correspondences and filter out correspondences according to the Lowe's ratio with a threshold of 0.8.

**Evaluation.** We adopt four evaluation metrics for comparison: inlier ratio, F-score, mean error, and median error. For each fundamental matrix hypothesis, we regard the correspondences whose reprojection error is less than 0.1 pixel as inliers. Inlier ratio is defined as the percentage of inlier correspondences among all input correspondences. F-score is the correspondences that are both inliers of the estimated model and the ground truth model, which denotes how well the estimated the model is aligned to the ground truth model. We also measure the epipolar error by first projecting the points in one image onto the corresponding epipolar line in another image with the fundamental matrix estimation and then computing the corresponding point-to-line distance as the epipolar error. We analyze the epipolar error for all the inlier correspondences by comparing the mean and median values.

Table 1: Fundamental Matrix Estimation.

|  | inlier ratio(%) | F-score | Mean Err. | Median Err. |
|---|---|---|---|---|
| RANSAC (5) | 21.85 | 13.84 | 0.35 | 0.32 |
| USAC (14) | 21.43 | 13.90 | 0.35 | 0.32 |
| Deep F (15) | 24.61 | 14.65 | 0.32 | 0.29 |
| NG-RANSAC (3) | 25.12 | 14.74 | 0.32 | 0.29 |
| R-SAC (Ours) | 26.74 | 14.87 | 0.33 | 0.31 |

**Comparison.** We show the performance compairson in Tab. 1. RANSAC (5) and USAC (14) are classical algorithms, which do not utilize neural network to capture prior geometric knowledge. Deep F (15), NG-RANSAC (3), and R-SAC (Ours) select observations for the model estimation with a learned neural network, which consistently ourperforms RANSAC and USAC. R-SAC achieves higher inlier ratio and F-score, which means the fundamental matrix estimated by R-SAC covers more inliers and is better aligned to the ground truth fundamental matrix. Our mean and median epipolar error are larger than Deep F and NG-RANSAC because the fundamental matrix estimated by R-SAC covers more inliers, which are most boundary observations. The extra boundary observations are not ourliers indeed so they can improve the model estimation robustness and accuracy.

## 5 Conclusion

We propose a reinforcement learning framework R-SAC for outlier rejection in model estimation. Based on the feasible environment design and the stop probability modeling, the agent can learn to generate a model covering as more inliers as possible to improve the model estimation accuracy with less iterations in a unsupervised manner. The experiment shows that our R-SAC achieves comparable performance to the state-of-the-art methods in fundamental matrix estimation.

## References

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
