# OpenReview forum: "R-SAC: Reinforcement Sample Consensus"
_CUHK.edu.hk/2021/Course/IERG5350_

### Official Review · AnonReviewer2 · 2020-12-18
**The work is good, satisfying the requirements of the project. All sections are described in concretely and originally. The work provides a good research direction.**

**Rating:** 7
**Confidence:** 4

**Review:**

Overall description: The work is good, satisfying the requirements of the project. All sections are described in concretely and originally. The work provides a good research direction.

Pros:
1.	The proposed approach is described with formulas and figure, helping the description clear.
2.	4 baseline methods and 4 evaluation methods are used to evaluate the performance of the proposed approach. The analysis of the experiment result is clear and easy to follow.
3.	The overall structure of the report is clear.

Cons:
1.	When describing the neural network architecture (section 3.3), suggest to add a figure illustrating the network architecture, which helps make things more clear and straightforward.
2.	Suggest to use fewer words to describe the classical algorithm and cut the abstract content, to make readers more focused on your work.
3.	Suggest to rewrite “ The extra boundary observations are not ourliers indeed so they can improve the model estimation robustness and accuracy.”in the comparison (Page 6), it is a little bit weird in the logic.
4.	Suggest to add the experiment setup part (add some details about settings about hyperparameters …)
5.	Suggest to add some description about Deep F method in the related work.
6.	In the abstract “ Empirical results show that our method achieves comparable performance compared with the previous supervised counterparts and remarkable efficiency especially when the outlier ratio is large.” It is a little bit hard to understand “large”, I suggest you to give more concrete numbers to describe it. But in the experiments there is no result showing when the outlier is small, the performance is good, when the outlier ratio is large, the proposed approach shows remarkable efficiency. Therefore, I suggest you to add more experiment results.
7.	Some small typos: “to approach such problem”(abstract), “some recent works ()”(page 2), “ourliers indeed”(Page 6).

---

### Official Review · AnonReviewer3 · 2020-12-20
**Novel RL adaptation to a classic problem**

**Rating:** 8
**Confidence:** 4

**Review:**

This work proposes reinforcement sample consensus (R-SAC) to train a neural network to classify the inliers and outliers
among all the correspondences with reinforcement learning, in particular the PPO algorithm. The number of inliers is the reward and thus the the agent operates in an unsupervised manner.
Personally I have very limited background on the topic of sample consensus, so the comments would be very general ones. The problem is well-studied and the problem definition is precise. The proposed approach is also novel as compared to the related work described, where this work amends the sampling strategy and the formulation as a self-supervised agent. The algorithm and the adaptation of PPO to the problem of sample consensus are described precisely in detail. Empirical results show that R-SAC can achieve performance comparable to the supervised counterparts. Improvement in efficiency is also observed. Overall, this work looks sound and promising.